# Coffee Oil Extraction Methods: A Review

**DOI:** 10.3390/foods13162601

**Published:** 2024-08-20

**Authors:** Raquel C. Ribeiro, Maria Fernanda S. Mota, Rodrigo M. V. Silva, Diana C. Silva, Fabio J. M. Novaes, Valdir F. da Veiga, Humberto R. Bizzo, Ricardo S. S. Teixeira, Claudia M. Rezende

**Affiliations:** 1Aroma Analysis Laboratory, Institute of Chemistry, Federal University of Rio de Janeiro, Rio de Janeiro 21941-909, Brazil; quelcoldibelli@gmail.com (R.C.R.); drigovelloso11@gmail.com (R.M.V.S.); 2Bioethanol Laboratory, Department of Biochemistry, Chemistry Institute, Federal University of Rio de Janeiro, Rio de Janeiro 21941-909, Brazil; sposina55@gmail.com; 3Department of Biochemistry, Institute of Chemistry, Federal University of Rio de Janeiro, Rio de Janeiro 21949-909, Brazil; mariafsmota@eq.ufrj.br; 4Chemistry Department, Federal University of Viçosa, Viçosa 36570-900, Brazil; diana.cardoso@ufv.br (D.C.S.); fabio.novaes@ufv.br (F.J.M.N.); 5Chemistry Section, Military Institute of Engineering, Rio de Janeiro 22290-270, Brazil; valdir.veiga@gmail.com; 6Embrapa Agroindústria de Alimentos, Rio de Janeiro 23020-470, Brazil; humberto.bizzo@embrapa.br

**Keywords:** Arabica coffee, Canephora coffee, green coffee oil, roasted coffee oil, Soxhlet, press, supercritical fluid

## Abstract

Green and roasted coffee oils are products rich in bioactive compounds, such as linoleic acid and the diterpenes cafestol and kahweol, being a potential ingredient for food and cosmetic industries. An overview of oil extraction techniques most applied for coffee beans and their influence on the oil composition is presented. Both green and roasted coffee oil extractions are highlighted. Pressing, Soxhlet, microwave, and supercritical fluid extraction were the most used techniques used for coffee oil extraction. Conventional Soxhlet is most used on a lab scale, while pressing is most used in industry. Supercritical fluid extraction has also been evaluated mainly due to the environmental approach. One of the highlighted activities in Brazilian agribusiness is the industrialization of oils due to their increasing use in the formulation of cosmetics, pharmaceuticals, and foods. Green coffee oil (raw bean) has desirable bioactive compounds, increasing the interest of private companies and research institutions in its extraction process to preserve the properties contained in the oils.

## 1. Introduction

Coffee is one of the most popular beverages and most traded commodities worldwide. It is cultivated in about 80 countries and, according to the USDA [1], the global production for 2024/2025 is estimated at 176.2 million bags of coffee (60 kg/bag), mainly due to the recovery of Brazil and Indonesia production. Brazil has been the leading producer and exporter of coffee, accounting for approximately 39% of world coffee production and is one of the most representative consumers. World coffee production is mainly focused on selling beans for hot beverages for domestic use across countries, as well as in bars, restaurants, and hotels. The unique flavor and aroma of roasted coffee makes it possible to obtain a very pleasant oil for the food industry, used in candies, chocolates, ready-to-eat drinks, for gourmet applications, and for instant coffee aromatization.

Coffee is also recognized for the production of oil rich in unsaponifiable matter, obtained from both green (or raw) and roasted beans, with quite distinct sensorial and physico-chemical aspects. Green coffee oil is greenish yellow with a slight odor usually obtained via pressing. Its main application is in cosmetics and skin care products due to its antioxidant and moisturizing properties as well as for the growing demand towards natural products. Roasted coffee oil is a brown viscous liquid usually obtained via pressing or CO_2_ supercritical extraction, the color and aroma of which are mainly related to the Maillard reaction that occurs during the roasting process. Chemically, aldehydes, ketones, furans, pyrazines, and other *N*-heterocycles, besides phenolic compounds, are the most representative volatiles [2].

Although there are 124 species of coffee (*Coffea* sp.) [3], only two are commercially important: *Coffea canephora* P. (about 25% of world production) and *C. arabica* L. (75%), with their diversity of botanical varieties. Among the coffee compounds, up to 17% correspond to a complex lipid fraction that gives rise to coffee oil from Arabica beans and up to 10% are observed in Canephora, all of them serving as important source of bioactive components [4,5,6]. The major class of compounds in coffee oil are triacyl glycerides (up to 75%). As reported by Folstar (1985) [7], the fatty acids of the coffee oil are present predominantly in two triacyl glycerides, the palmitoyl-linoleoyl-palmitoyl (about 28.1%) and palmitoyl-linoleoyl-linoleoyl (about 27.5%), with significant quantities of stearoyl-linoleoyl-palmitoyl (about 8.6%), linoleoyl-linoleoyl-linoleoyl (about 6.7%), palmitoyl-oleoyl-palmitoy (5.9%), and stearoyl-linoleoyl-linoleoyl (4.2%).

In Arabica beans, there is also a representative amount of diterpene esters (around 17%) besides free and esterified sterols, serotonin amides, phosphatides, and tocopherols, as will be discussed later. An important difference in the constitution of coffee oil from other beans is the presence of the esterified diterpenes cafestol and kahweol, which are not found in any other matrix [6]. Related to the green beans, the levels of total phenolic compounds vary between 4% and 8.4% for *C. arabica*, while for *C. canephora* it is between 7% and 14.4% [8]. Differences in alkaloid concentrations between *C. arabica* and *C. canephora* are significant, with the latter generally having higher levels of caffeine (around 1% for Arabica and up to 2.7% for Canephora) and lower amounts of trigonelline [9,10,11].

One of the highlighted activities in the Brazilian agribusiness is the industrialization of oils due to their increasing use in the formulation of cosmetics, pharmaceuticals, and foods [12]. Green coffee oil (raw bean) has desirable bioactive compounds that increase the interest of private companies and research institutions in its extraction process to preserve the properties contained in the oils. Several methods for coffee oil extraction are proposed to increase their lipid content. In the literature, physical and chemical treatments are applied to the beans, and their parameters are usually discussed in terms of efficiency, applicability, cost, and environmental risks [13].

Some advanced extraction technologies have emerged since conventional Soxhlet and pressing extraction techniques, such as microwave-assisted enzymatic extraction, ultrasound-assisted extraction, supercritical fluid technology, high-pressure-assisted extraction, and pulse electric field-assisted extraction (Figure 1), ranging from the yield of oil to aspects that impact the sustainability of the process [14,15]. Brazil also has the world’s largest coffee research program through the Brazilian Coffee Research and Development Consortium—CBP&D/Café—coordinated by the Brazilian Agricultural Research Corporation Embrapa, with research investments in genetic improvement, pest management, irrigation, production quality, and biotechnology, with concern for economic sustainability and environmental preservation [16].

Considering this context, alongside Brazil’s significant role in the agricultural coffee sector, this article aims to discuss aspects related to the most-used processes for coffee oil production, as well as the chemical composition of green and roasted coffee oils. Soxhlet, pressing, microwave-assisted extraction, supercritical fluid extraction and more recently, extrusion associated to Soxhlet were the techniques covered in this text.

## 2. Green and Roasted Coffee Oils

### 2.1. Chemical Composition of Green Coffee Oil

Lipids of green coffee beans are located in the endosperm and to a lower extent as a wax on the outside of the bean. The lipid fraction in Arabica coffees is around 17%, while Canephora coffee content is usually lower, up to 10% [6,17].

Arabica and Canephora coffee beans are composed of triacylglycerols—TAG, esterified *ent*-kaurane diterpenes, free diterpenes, sterols and their esters, tocopherols, phospholipids, and serotonin amides [18,19,20]. Lipids of *C. canephora* var. robusta coffee were recently detailed and quantified via lipidomics through LC-ESI-MS/MS [21]. However, this study did not quantify sterols, tocopherols, and diterpene esters. In order to illustrate the difference in composition of Arabica and Canephora coffee lipids, Figure 2A,B was constructed based on data from Lercker et al. [18].

Speer and Kolling-Speer (2006) detailed some differences between Arabica and Robusta coffee oils and verified that stearic acid content (in esterified form) is much lower than that of oleic acid for Robusta samples, while in Arabica both are present in equivalent quantities. Also, the total amount of diterpenes in green Arabica coffee varies from 1.3% to 1.9% (*w*/*w*), while in Robusta beans it ranges from 0.2% to 1.5% (*w*/*w*) [22].

Arabica coffees diterpenes are mainly composed of cafestol (2.99–5.84 g/kg) and kahweol (4.15–6.73 g/kg) and small amounts of 16-*O*-methylcafestol (0.01–0.14 g/kg); the chemical structures are represented in Figure 3. Robusta coffees have 16-*O*-methylcafestol (0.45–1.39 g/kg), cafestol (0.76–1.91 g/kg), and small amounts of kahweol (0.04–0.12 g/kg) [23]. 16-*O*-MC is one of the major markers for detecting the presence of Robusta coffee in blends [7]. Cafestol and kahweol are vastly explored in the scientific literature [6,24,25,26] and have been correlated to antitumor, antioxidant, and anti-inflammatory properties. Cafestol is known to present a hypercholesterolemic effect, and many studies have already associated this biological activity to the ingestion of non-filtered hot beverages as in Turkish and French Press Arabica brews [27,28,29].

The main fatty acids present in green coffee beans are linoleic (C18:2) and palmitic (16:0) acids, ranging from 42.5 to 50.2% for C16:0, 8.9 to 17.5% for C18:0 (stearic acid), 7.2 to 11.0% for C18:1 (oleic acid), 20.5 to 25.8 for C18:2, 4.5 to 6.0% for C20:0 (arachidic acid), and 1.1 to 2.5% for C22:0 (behenic acid) [4,6]. Besides acylating mono-, di-, and triacylglycerides in coffee beans, these acyl moieties also appear as esterifying diterpenes, sterols, serotonin, and minor lipids [6,30,31,32].

The sterol content in the coffee lipid fraction is very similar to other bean vegetable matrices. β-Sitosterol (52%), stigmasterol (22%), and campesterol (16%) represent nearly 90% of this class, while 5-avenasterol, campestanol, 24-methylenecholesterol, sitostanol, 7-stigmastenol, 7-avenasterol, 7-campesterol, and clerosterol represents the remaining 10% [6,33,34].

The coffee oil lipid unsaponifiable fraction also contains ^β^*N*-alkanoyl-5-hydroxytryptamines (Cn-5HTs), as seen in Figure 4, which consist of a serotonin unit conjugated to acyl moieties through an amide link, mostly represented by C-20 and C-22 units [32]. Cn-5HTs, typically found in the wax bean, have interesting biological properties, such as anti-inflammatory, antinociceptive, anxiolytic, and others [32,35,36,37], but also are correlated to stomach irritation [38,39]. Due to this property, associated to the hypercholesterolemic effect of cafestol, a fermentative wet post-harvest process with commercial yeasts with *Saccharomyces cerevisiae* was developed in which substantial reductions for C-5HTs (up to 38% for C20-5HT and 26% for C22-5HT) as well as for diterpenes (54% for cafestol and 53% for kahweol) are achieved [40].

The cosmetic industry is interested in moisture retention in the skin and green coffee oil is also a potential ingredient to prevent photoaging [41,42]. Coffee oil provides properties against UV-B absorption without cytotoxic effects, being useful in sunscreen factor formulations and improving its stability, as well as for other cosmetic products [43,44,45,46,47]. Some of these properties may also be present in roasted coffee oil, as few components are altered by roasting profiles [12,48].

### 2.2. Chemical Composition of Roasted Coffee Oil

The roasting process consists of heating the green coffee beans to high temperatures (often exceeding 200 °C) to develop aroma and flavor components of interest, typically appreciated with roasted coffees. The time and temperature of the roasting depend on the desired characteristics of the final products [49,50,51]. In the roasting process, the coffee beans lose water, some volatile compounds are formed, and some are lost. Moreover, the degradation of carbohydrates, amino acids, and chlorogenic acids yield aroma compounds, mainly via Maillard and Strecker reactions, as well as melanoidins, brown nitrogen-containing polymers that accounts for up to 25% in the roasted coffee beans [49,50,51,52]. In roasted Arabica coffee, high amounts of fatty acids were observed, the main being linoleic acid (40.3%) and palmitic acid (34.5%) [12]. The lipid content in roasted coffee beans increases when compared to green beans since bean cell walls are partially degraded and allow the release of oil, which favors aroma retention. In general, the roasted Arabica coffee beans present about 4.1–16.8% of lipids, whereas the Canephora has 3.2–11.5% [53].

Roasted coffee oil is rich in diterpenes (3.72%), with a content of kahweol and cafestol of about 1.98 and 1.74%, respectively, depending on the roasting conditions. It also presents tocopherols (0.91%), with β-tocopherol (0.88%) as the main component, followed by α (0.03%) and δ isomers (0.002%). Moreover, this oil may also have chlorogenic acid (0.010%) and caffeine (0.35%), depending on the polarity of the solvent used in the extraction process [53].

Roasting temperature plays a fundamental role in modifying the coffee compounds, directly influencing the quality and properties of the bean [50]. The more intense the roast, the greater the fragility of the bean wall, and, consequently, more extravasation of the intracellular content occurs to the surface of the roasted bean. A recent study showed that in light roasting, there is a significant increase in the levels of cafestol and kahweol (C&K) compared to the raw bean, in addition to the appearance of seven thermal degradation products. In medium and dark roasts, nine more compounds were observed, with 10 derivatives from kahweol and 6 from cafestol [54]. Up to now, there are no studies of biological activities on these products formed via roasting.

A small amount of trans-fatty acids and peroxides are formed during coffee roasting [55]. The presence of trans-fatty acids is undesirable for food formulation due to its intake association with an increased risk of cardiovascular disease [56]. Peroxides also can negatively impact the taste of coffee, introducing bitter and metallic notes that compromise the sensory quality of the drink [57]. Peroxides are known for their ability to cause oxidative stress in the body, which can result in cell damage and contribute to the development of chronic diseases such as cancer and neurodegenerative diseases [58].

Another interesting study was performed by Guatemala-Morales et al. [59] related to the quantification of polycyclic aromatic hydrocarbons (PAHs) in roasted Arabica coffee. These substances have genotoxic properties and show risks to human health and, in coffee, are formed during the roasting of the beans. The authors validated a methodology for 16 PAHs via gas chromatography–mass spectrometry and found that roasting in a heterogeneous spouted bed reactor yielded a sum of the PAHs ranging from 3.5 to 16.4 µg kg^−1^. The sum of the PAHs indicated that their concentration rose with increasing roasting temperature.

Serotonin amides suffer an intense degradation during roasting, and an α-cleavage to serotonin was described, following the production of 5-hydroxyindole, 3-methyl-5-hydroxyindole, 3-ethyl-5-hydroxyindole, free fatty acids, amides, and nitrile compounds [55].

The roasted coffee oil contains most of the compounds responsible for the pleasant roasted coffee aroma; therefore, it is used to flavor products such as instant (soluble) coffee, cakes, and candies. This oil is also suitable for food formulation as a source of bioactive compounds (such as linoleic acid, tocopherols, and diterpenes) and for cosmetics applications due to its sun protection factor [53].

## 3. Coffee Oil Extraction Methods

The botanical species, bean quality, particle size, extraction time, and solvent, if present, are important parameters in coffee oil yield [6,60]. Several techniques have been used to obtain coffee oil, mainly pressing, Soxhlet extraction, microwave, and supercritical fluid, and important aspects related to the production process of oil recovery will be discussed below.

### 3.1. Mechanical Extraction

Extraction conditions generally impact the chemical composition and biological properties of the extracted oil. Extraction via mechanical pressing does not use organic solvents and, therefore, does not produce potentially toxic residues. It reduces environmental and human health concerns and so is attractive for the food and cosmetics industries.

Pressing is one of the oldest processes for extracting oils. Studies indicate that it was used in ancient times, employing stone mills powered with animal traction. It is a physical process in which oils are obtained from a range of matrices, especially oilseeds, and is usually used when there is interest in keeping original organoleptic properties. Production costs are reduced since no organic solvent is used. The total yield obtained through this process is usually inferior to solvent extraction (usually up to 60–80%), but in the case of coffee, lower yields are usually obtained. Temperature is often used to increase pressing yield when there are no thermolabile substances. Other parameters such as the seeds’ moisture content, screw speed, press exit size, and particle size can also impact the total yield [61,62].

There are two main types of presses used nowadays: hydraulic and screw press. The oil extraction process currently uses continuous presses (expeller), where the beans (for example, coffee) are moved forward and compressed with a worm screw. At the exit, an adjustable cone allows the pressure inside the press to be increased or decreased, resulting in crude vegetable oil and a cake, the solid part resulting from pressing. The oil must pass through a filter press to remove suspended solid parts, and the cake can be toasted, ground, and sold for consumption in beverage preparation [63]. In 2020, Hussein briefly reviewed some research on cold pressed green coffee oil focusing on its different applications including cosmetic products [64].

In general, the oil extraction capacity of raw coffee beans using mechanical presses tends to be lower compared to roasted beans, with yields varying between 6% and 10% depending on the specific pressing conditions [61,64,65,66].

#### Expeller Press

An expeller press is composed of a barrel shaped outer casing with perforated walls and a metal screw that feeds oil bearing products. The material is continuously grinded and crushed, rupturing the oil cells which turns possible oil collection through the perforations in the casing. The material from which oil is extracted is known as cake. Capacities may vary, but most expellers are able to process between 8 and 45 kg per hour [67].

Aiming to determine green and roasted defective coffee beans’ proximate composition and fatty acid profile, the beans were subjected to two subsequent pressing cycles in a Mazziero Press. Healthy coffee beans had a similar but higher oil content (10.8 ± 0.3%) compared to defective coffee beans (9.9 ± 0.1%). The fatty acid composition showed no significant difference between healthy and defective coffee beans. Screw-pressed unsaponifiable matter (12.8 g/100 g oil) and solvent-extracted PVA (9.2 g/100 g oil) oils showed significant differences, reinforcing that screw pressing is not a selective process to lipids [68].

The production of bioactive carboxymethylcellulose (CMC) films enriched with green coffee oil and its residues was investigated, where coffee oil was obtained by using a continuous expeller press at 25 °C. The temperature of the oil at the outlet of the expeller, measured with a thermocouple, varied between 40 and 45 °C. After the crude oil (oil with sludge) was centrifuged for 20 min at 3100 g and 25 °C, the oil content was 4.56 ± 0.74%. The phenolic compounds extracted from the pressing residues were incorporated into bioactive carboxymethyl cellulose films, managing to produce films with UV-radiation blocking properties and high antioxidant capacity. The CMC-based films with the highest antioxidant capacity were obtained by incorporating green coffee oil and a 40% hydroalcoholic extract of pomace, emulsified with lecithin. As a conclusion to the work, the authors report that the films developed, especially those containing green coffee cake, appear to be promising natural additives and for applications in active food packaging [69].

Silva et al. [61] studied how some pressing parameters affect the yield and lipid composition of green Arabica coffee oil, focusing on fatty acids, diterpenes, and serotonin amides content, with the latter being described for the first time in the context of a coffee oil composition. A design of experiments was made to evaluate particle size (<0.850 mm; >0.850 mm; and <2.00 mm), screw speed (18 and 30 rpm), press exit nozzle size (4 and 5 mm), and preheating (on and off). Oil yield varied between 2.65 and 6.27% and all the parameters had a significant impact on this yield. Diterpenes kahweol and cafestol content ranged from 13.33 to 16.72 mg/g and 37.11 to 47.14 mg/g of oil, respectively. Serotonin amides content ranged from 114.42 to 577.37 µg/g for C20-5HT and 193.50 to 1068.08 µg/g for C22-5HT. The difference in fatty acids composition related to press conditions was not significant. In this case, and in the study of Vidal at al [69], it a laboratory screw expeller press with a capacity of 1 kg/h was used.

In conclusion, mechanical pressing is a natural and straightforward approach that does not use chemical solvents, offering environmental and quality advantages when compared to solvent extraction methods that use organic solvents, avoiding the solvent recovering step and possible problems with solvent residue and pollution, making it attractive to the food and pharmaceutical industry when seeking high-quality and less-processed products. However, it tends to offer a lower yield and cannot achieve the same oil purity obtained with more advanced methods [70,71].

### 3.2. Soxhlet Extraction of Coffee Oil and Its Comparison to Other Techniques

In this topic, articles that deal with coffee bean extraction via Soxhlet alone or combined with other techniques will be reviewed. Supercritical fluid will be briefly compared here and discussed in detail in Section 3.4, as more detailed studies for coffee were performed with this technique.

Soxhlet extraction is one of the official standard methods to determinate oil content [72]. In summary, the sample is placed on a cellulose filter (thimble) that is gradually filled with solvent (also called fresh condensed extract) from a distillation flask. When the liquid reaches the overflow level, a siphon sucks it from the thimble and discharges it back into the distillation flask, thus transporting the extracted analytes into the bulk liquid. This operation can be repeated until complete extraction is achieved.

Speer and Kolling-Speer [6] cited two official methods for oil extraction via Soxhlet, the German Foundation for Scientific Research (Deutsch Forschungsgemeinschaft, DFG, 1952) [73] and the AOAC (2005) [74]. Both use petroleum ether for 4 and 16 h, respectively. They also cite the study by Picard et al. [75], who emphasized that different solvents, such as diethyl ether, petroleum ether, hexane, and their mixtures make results comparison difficult, a problem that remains today.

In an attempt to summarize the methods that will be discussed next and allow the reader to better compare the conditions used and the results obtained, Appendix A with this information can be consulted in the Appendix A.

Soxhlet extraction was used to obtain Arabica and Canephora coffee oils that were analyzed for their sterol content. About 50 g of ground beans were extracted with hexane for 8 h, siphoning six times per hour. The oil yield for Canephora coffees was 12–17% and for Arabica 15–20%. Using sterols as chemical descriptors, Δ 5-avenasterol and sitostanol were considered the most differentiating variables. Based on these descriptors, the authors propose that any new coffee sample could be easily classified [76]. A significantly higher oil content was described in this article when compared to others, as will be discussed. It is important to note that this study broaches an important parameter in Soxhlet extraction studies, which is the number of siphonages. Most articles that use Soxhlet report the extraction time but do not reference the number of siphons used (as can be seen in Appendix A), this being the moment where the extraction occurs, so it is an important parameter to be measured.

Green and roasted Arabica coffees were submitted to Soxhlet extraction, performed using hexane (10 mL/g sample) for 16 h, and compared to supercritical fluid extraction (SFE). In SFE, the authors investigated temperature, type of material, particle size and type of co-solvent (ethanol, acetone, and ethyl acetate) in a fixed ratio of solvent to solid mass (5:1). The oil yield and the diterpenes were determined and compared. Green coffee oil content was 11.37% and 15.49% for roasted beans via Soxhlet, while for diterpenes it was 3.84 g/kg of cafestol and 4.76 g/kg of kahweol for the green beans and 3.28 g/kg and 3.98 g/kg for the roasted beans, respectively [77]. Supercritical data will be discussed, but the authors showed that the content of the diterpenes in the oils obtained from Soxhlet was higher than SFE.

The recovery of oil from healthy and defective Arabica coffee beans using an industrial Soxhlet device, with a capacity of 25 kg, and hexane for 16 h was investigated for biodiesel application. Solvent was removed in a rotary evaporator and a transesterification step was carried out. The Soxhlet method’s oil yield varied from 10 to 12%, and the highest ester yield (70.1% for healthy beans and 73.8% for deficient beans) was obtained with MeOH at 25 °C for 1 h [78]. Regardless of the ester yields, coffee oil showed potential as a candidate raw material in biodiesel production.

The roasting condition of Robusta coffee beans was studied focusing on the influence on coffee oil properties. Oil content, fatty acids composition, including *trans*-fatty acids, peroxide value, and conjugated dienes and trienes were measured, as was the bean aroma. Robusta coffee beans were roasted by varying roasting temperature (190 to 216 °C), air speed (0.5 and 1 m/s), and air humidity (0.07 to 1%). Oil recovery took place in an automatic Soxhlet apparatus for 1 h using petroleum ether at 40 to 60 °C, and the solvent was recovered in the apparatus itself. The best oil recovery conditions were at a roasting temperature of 216 °C, air flow speed of 0.5 m/s, and dry air, with an oil yield of 11.31%; at 210 °C, 1.0 m/s, and humid air, the oil yield was of 11.20%; and at 216 °C, 1.0 m/s, and dry air, the oil yield was of 11.07%. The first two conditions resulted in enriched aroma oils [55].

Diterpenes from roasted Arabica coffees extracted with the Bligh Dyer method and cold and hot saponification were compared to Soxhlet for 6 h with *t*-butyl methyl ether. Saponification with KOH was done to obtain the unesterified diterpenes. An apparent overestimation of oil extracted via the Soxhlet method was observed (18.6% yield) compared to Bligh Dyer (13.5%). For the diterpenes, the Bligh Dyer method proved to be less efficient than the Soxhlet method, whose contents of cafestol and kahweol corresponded to 170.2 and 318.7 mg/100 g, respectively [79].

Another interesting technique is microwave-assisted extraction (MAE), which has already proved to be efficient for oil bean recovery [80] and was compared to Soxhlet for coffee oil extraction.

Green Arabica coffee oil extraction via microwave-assisted extraction (MAE) and Soxhlet using petroleum ether showed a coffee oil content varying from 7.5 to 9.5% via the Soxhlet, while with MAE, there was a variation from 5.8 to 7.6%. Quantification of cafestol and kahweol diterpenes was monitored via HPLC/UV. A full factorial design was applied for the MAE to evaluate time (2, 6, and 10 min) and temperature (30, 37.5, and 45 °C) parameters. The best condition for MAE was achieved at 45 °C and 10 min, which was much faster when compared to a 4 h Soxhlet extraction. Regarding the cafestol and kahweol content, green coffee oil obtained via MAE could lead to a space–time diterpenes yield six-times higher when compared to the Soxhlet method. Another advantage of MAE is the reduced amount of solvent needed when compared to the traditional Soxhlet extraction method [81].

Microwave-assisted extraction (MAE) applies microwave energy to heat the solvent in contact with a sample matrix to extract target compounds. Under conventional heating, the sample in contact with the equipment wall is first heated, thus heating occurs from the external to internal environment with heat transfer via conduction, followed by radiation and convection transfer. Heating via microwave irradiation will depend on the solvent’s dielectric properties and usually takes about 15–30 min and uses a solvent volume of around 10–30 mL, which represents a considerable extraction time and solvent volume reduction compared to the Soxhlet extraction method [82].

Green Arabica coffee oil was also obtained via Soxhlet and supercritical extraction, which showed a similar chemical composition according to ATR-FTIR infrared spectra. Soxhlet was performed for 7 h using hexane and supercritical CO_2_ extraction, and the yield was higher and required less extraction time than Soxhlet, as will be detailed later [83].

Oliveira et al. [84] evaluated the role of different solvents (acetone, ethanol, ethyl acetate, hexane, isopropanol, and petroleum ether) in Soxhlet extraction related to total soluble solids and some bioactive compounds from green coffee beans. The soluble solids/oil extraction was performed via Soxhlet for 3 and 5 h and the temperature was set according to the boiling point of each solvent. As a result, yield of soluble solids/oil content was 8.31% using acetone, 11.78% using ethanol, 6.44% using ethyl acetate, 8.85% using hexane, 10.23% using isopropanol, and 7.67% using petroleum ether. Ethanol was found to be the ideal solvent for extractable phenolic and antioxidant compounds. Only for β-carotene bleaching assay (BCBA) analysis did ethyl acetate yield the best results for antioxidant activity.

Recently, Ribeiro et al. [85] used extrusion pretreatment to extract coffee oil from healthy and defective beans followed by Soxhlet. The study varied the temperature parameters (40 to 80 °C) and screw rotation speed (60 to 100 RPM) for extrusion, and Soxhlet extraction consisted of using 20 g of material and 150 mL of hexane for 4 h. The optimized extrusion condition (68 °C and 60 rpm) with subsequent Soxhlet extraction resulted in oil contents of 16.42% and 15.29% for healthy and defective grains, respectively. Extrusion pre-treatment oil levels were significantly higher than those found for only Soxhlet, being 9.05% and 9.47% for healthy and defective grains, respectively. The extruder’s ability to deconstruct the raw coffee bean recalcitrant structure and consequently the lipid pockets was the main cause associated with this excellent performance.

Soxhlet, in comparison to other extraction techniques, usually has disadvantages such as the long extraction time (4–16 h) and the large amount of solvent used, producing more waste. Another disadvantage is the exposure of the extracted solute to the solvent boiling point for a long period of time and the possibility of heating thermolabile compounds, bringing undesirable results [86,87,88].

There are still few studies that explore the scale-up of Soxhlet for industrial applications. Soxhlet transition to large-scale extraction processes faces significant challenges, such as the need for larger volumes of solvents, temperature and pressure control, as well as energy efficiency and safety issues. Therefore, while Soxhlet continues to be the method of choice for detailed and accurate analyses in the laboratory, application on an industrial scale still requires further research and development to overcome these obstacles and enable its use in larger extraction processes [89,90].

### 3.3. Supercritical Fluid Extraction (SFE)

Supercritical fluid extraction (SFE) is used to extract components from different matrices using a fluid in temperature and pressure conditions above its critical point. Supercritical fluid forms a homogeneous phase that presents both liquid-like and gas-like properties. Physicochemical properties such as viscosity, diffusivity, and dielectric constant can be controlled by varying the supercritical fluid temperature and pressure, without phase changes occurring [91,92]. These properties provide supercritical fluids with solvation power similar to liquids, acting as solvents. High diffusivity and low viscosity allow them to have properties like gases, with high penetration power in solid matrices, favoring extraction process mass transfer [93,94].

SFE is considered a selective extraction that presents low environmental impact when compared to solvent extraction, as SFE allows high extraction yields with no or small amounts of organic solvent. Moreover, SFE requires a low extraction time and is inexpensive to operate. However, the high setup cost is a disadvantage when compared to other extraction techniques [95,96].

There are different substances applied for supercritical extraction, such as carbon dioxide, ethanol, propane, and water, for example. Among these, carbon dioxide is the most used due to its availability at high purity, non-toxicity, non-flammability, low cost, and relatively low critical temperature (31.1 °C) and pressure (72.8 bar) in comparison to other supercritical fluids [97,98]. However, CO_2_ is a nonpolar solvent and, therefore, its affinity is limited. To overcome this disadvantage, a co-solvent may be applied to increase supercritical CO_2_ solvation power and, consequently, increased polar analytes solubility. Ethanol and methanol are the most common co-solvents applied [96].

CO_2_ supercritical fluid extraction with and without the addition of co-solvents has been extensively applied for green, roasted, and spent coffee oil extraction for food, pharmaceutical, and cosmetic applications, as can be seen in the next paragraphs. Supercritical fluid extraction presents itself as an attractive technique to obtain raw and roasted coffee oils due to the ease of recovering the solute (coffee oil) and recycling the solvent (CO_2_) via simple thermodynamic control of pressure and temperature [92].

Commercial green and roasted coffee extraction was studied with CO_2_ SFE by optimizing temperature (60–90 °C) and pressure (235–380 bar) conditions. The oil yield and diterpene levels were compared with the Soxhlet extraction with hexane, as previously discussed. In general, the correlation between extracted oil yield and diterpene content was inverse. CO_2_ supercritical green coffee oil’s most efficient extraction condition was achieved at a temperature of 90 °C, pressure of 373 bar, and CO_2_ density of 0.77 g/mL. However, this condition presented the lowest diterpenes concentration (4.14 g/kg). The CO_2_ supercritical extraction that presented the highest concentration of diterpenes (0.45 g/kg) was at 70 °C, 327 bar, and with CO_2_ at 0.81 g/mL, a content 43% lower when compared to oil obtained with Soxhlet extraction. Regarding the roasted coffee, this oil presented a significantly lower content of diterpenes when compared to green coffee oil, showing a reduction of 14.4% and 16.5% for cafestol and kahweol, respectively. The best oil extraction condition was at 70 °C, 371 bar, and CO_2_ at 0.84 g/mL, a condition that presented the lowest diterpenes concentration (0.21 g/kg) [77].

Green coffee oil, caffeine, and chlorogenic acid extraction was investigated using CO_2_ SFE and CO_2_ SFE with the addition of ethanol (5% *w*/*w*) and isopropyl alcohol (5% *w*/*w*) at 50 and 60 °C and 152 and 352 bar. The Soxhlet extraction using benzene, a very toxic solvent, was carried out to estimate the total oil content. An increase in the pressure resulted in higher oil extraction yield with all the solvents tested (pure CO_2_, CO_2_–ethanol, and CO_2_–isopropyl alcohol). The increase in extraction temperature decreased the oil yield when using only CO_2_ as a solvent. However, with the use of a co-solvent, a different behavior was observed, and the increase in temperature resulted in a higher oil extraction. Under the same process conditions, the oil extraction was higher when using CO_2_–ethanol, followed by CO_2_–isopropyl alcohol. The CO_2_ conditions at 60 °C and 352 bar provided an extraction of 17.65 g of the oil mass fed. Regarding caffeine and chlorogenic acid extraction, these compounds showed less affinity when compared with the coffee oil, and low yields were achieved [97].

CO_2_ SFE was also optimized to obtain a green coffee oil enriched in diterpenes, cafestol, and kahweol. The oil extracted at 200 bar and 70 °C presented a cafestol content of 50.2 and kahweol at 63.8 mg/kg green coffee oil. This value was higher than that achieved via conventional pressing methods which presented yields of 7.5 and 12.8 mg/kg green coffee oil for cafestol and kahweol, respectively. Regarding the fatty acid composition, the values obtained with the supercritical fluid were in agreement with the literature, presenting linoleic acid (38.3%) and palmitic acid (32.4%) as the major fatty acids [65].

The roasted coffee oil from CO_2_ SFE was also evaluated with a central composite experimental design, performed to establish the effect of the pressure (150–300 bar) and temperature (40–60 °C) on the oil yield and fatty acid composition. Response surface analysis indicated that pressure had a greater influence on oil extraction yield than temperature, and the optimum yield (8.9%) was obtained at 331 bar and 35.9 °C. Regarding the fatty acids profile, all extraction conditions showed palmitic and linoleic acid as the major fatty acids. The optimum linoleic acid extraction (37.8%) was obtained at the same condition as the optimum yield. However, the optimum palmitic acid extraction (50.3%) was obtained by increasing the temperature to 64.1 °C [98].

CO_2_ SFE green coffee oil solubility was studied via a static method under temperatures and pressures ranging from 40 to 80 °C and 300 to 350 bar, respectively. At 300 bar, green coffee oil solubility increased with temperature, between the range of 40 and 60 °C. However, at 70 °C and 80 °C the solubility decreased. Similar results were observed at 350 bar, with an increasing in solubility with the increase in temperature to 70 °C, but decreasing at 80 °C. Regarding the yields, the highest ones were achieved at 70 °C and 300 bar (7.58%) and 80 °C and 350 bar (7.60%), while the content obtained via Soxhlet was 7.57%. There was no significant difference between the fatty acid profile obtained via Soxhlet extraction and the different experimental conditions with the supercritical CO_2_ extraction [99].

CO_2_ SFE green coffee oil solubility at different temperatures (40–60 °C), pressures (200–400 bar), and supercritical CO_2_/ethanol ratios (0–5.7% *w*/*w* of ethanol) was also investigated. The green coffee oil solubility values increased at higher pressures. By using 2.9% of ethanol, the solubility was 63% higher than the one in pure supercritical CO_2_ and the crossover pressure point was about 20 bar higher. The extract phase obtained with supercritical CO_2_ and supercritical CO_2_/ ethanol presents fatty acids, kahweol, and cafestol contents up to 3.4-, 4.4-, and 4.0-times greater than the green coffee oil extracted via pressing. Regarding the fatty acids content, the mass percentage of each was similar between the green coffee oil extract via pressing, supercritical CO_2_, and supercritical CO_2_/ethanol [100].

The green coffee oil CO_2_ SFE using ethanol as co-solvent was also evaluated for temperatures ranging from 50 to 70 °C, pressure ranging from 15.0 to 30.0, and ethanol content ranging from 5 to 20%, for the green coffee oil yield and total phenolic compound content using a face-centered central composite design. The pressure and co-solvent content showed a positive impact on the extraction yield. Regarding the total phenolic content, the co-solvent presented a positive impact, while the temperature presented a maximum yield at 62 °C. The experimental data were fitted to a second-order polynomial model and the desirability suggested that the optimal conditions were at 300 bar, 62 °C, and 20% of co-solvent, predicting an extraction yield of 7.7% and a total phenol content of 5.4 mg gallic acid equivalent per g of green coffee supercritical extract (GCSE). Moreover, the effect of the temperature on the caffeine and 5-caffeoylquinic acid content was also evaluated by comparing the values at the optimal condition with those extracted at 20% co-solvent, 300 bar, and 50 °C. The extract obtained for the optimal condition showed higher content of caffeine and 5-caffeoylquinic acid, and, according to the authors, this behavior is related to a decrease in the density of the solvents at a higher temperature [101].

Supercritical fluid extraction is a green technique for extracting target compounds due to the non-use of toxic solvents that can pollute the environment. The properties of supercritical fluids allow the extraction to be carried out at relatively low temperatures in the absence of oxygen and light, avoiding thermo- and photodegradation of sensitive compounds, as well as oxidation reactions that degrade the sample [15,86].

Therefore, although it requires a high cost in terms of equipment and complex processing, SFE makes it possible to obtain a high purity oil, without solvent residues, and is a highly efficient and sustainable method, preserving the bioactive compounds, offering advantages for the food and pharmaceutical industries which require high oil quality [102].

### 3.4. Other Techniques

Other authors compared ultrasonic/microwave-assisted extraction (UMAE), microwave-assisted extraction (MAE), ultrasound-assisted extraction (UAE), and pressurized liquid extraction (PLE) for the green coffee oil yield and composition. The ultrasound technique consisted of using a working frequency of 40 kHz and electrical power of 50 W, at a sample:ethanol ratio of 1:30 (g/mL) at 35 °C for 50 min. The microwave technique consisted of using 10 g of powdered sample with 100 mL of ethanol extracted at 60 °C for 30 min, and the microwave power was set at 200 W. The ultrasonic-microwave-assisted extraction used a mass ratio with ethanol of 1:28 (g/mL) at 60 °C for 10 min, with a microwave power of 350 W. The last technique, pressurized liquid extraction, consisted of using a sample-to-ethanol ratio of 1:2 (g/mL) at 100 °C for 30 min, and the pressure reached 100 bar. Among the four techniques evaluated, the ultrasonic/microwave-assisted extraction presented the highest yield (10.58%), followed by microwave-assisted extraction (9.34%), ultrasonic-assisted extraction (9.06%), and pressurized liquid extraction (6.34%). The ultrasonic/microwave-assisted extraction presented the highest yield (10.58%), while pressurized liquid extraction (6.34%) was the lowest. The green coffee oil extract from pressurized liquid extraction showed the highest tocopherol, total phenolic compounds, and phytosterol content, while the oil extracted via the ultrasonic/microwave-assisted extraction showed the highest diterpenes content. Regarding the fatty acid content in the four green coffee oils evaluated, the different techniques did not show an overall effect, but the ultrasonic/microwave-assisted extraction showed a subtle decrease in the linoleic acid content and an increase in palmitic acid [103].

A mechanical hydraulic press, much less used than an expeller press, was used to investigate the effect of pressure applied through a flat piston on wet- and dry-roasted defective coffee beans (RDCBs) aiming for lipid recovery. The RDCBs were pressed at room temperature for 60 min, with the pressure applied increasing from 350 to 550 bar at 20 min intervals. As a result, lipid amount corresponded to a yield of 2.47% *w*/*w*, or a recovery of crude lipids of 21.6% *w*/*w* in relation to the average yield in lipids extracted via Soxhlet with hexane (11.41%) [104]. Due to the low oil recovery through the hydraulic press, this method is not very popular in coffee oil extraction. Although its use is declining, hydraulic pressing is still used for processes such as cocoa butter extraction, virgin olive oil production, and oil production from some other matrices [104,105].

To establish a basis for the generation of related patents, a preliminary search was conducted in the World Intellectual Property Organization (WIPO) database using the terms “coffee oil” AND “extraction”. This research resulted in the identification of 61 documents, of which 16 are associated with the United Kingdom, 8 with Japan and the United States, 5 with the Russian Federation, 4 with Canada and the Republic of Korea, 3 with New Zealand and the Cooperation Treaty in Patents (PCT), and 2 to China and the European Patent Office. The documents were published between 2015 and 2024. However, given that these patents are not the main focus of this work, they will not be analyzed in greater depth.

## 4. Prospects for Extraction Processes

In recent years, coffee oil extraction has seen significant advances, with a focus on process optimization and sustainability. Advanced extraction methods, such as supercritical fluid extraction, are gaining prominence as they offer greater efficiency reducing the use of organic solvents [106].

Coffee oil is also being explored in new applications, such as biodegradable films and sustainable packaging, due to its antioxidant properties, demonstrating the potential of coffee oil to improve the functionality and sustainability of packaging materials [69].

As sustainability has become a priority, there has been a growing interest in using coffee waste, such as pomace and grounds, to extract oil and other compounds. This approach not only reduces waste, but also creates value from by-products [107].

## 5. Conclusions

Coffee is one of the most traded commodities in the world, and its composition presents a considerable amount of lipids. Coffee oil is an important source of bioactive compounds, including diterpenes, tocopherols, and unsaturated fatty acids such as linoleic acid. Both green and roasted coffee oils have potential use in the cosmetic industry in sunscreen formulations. In addition, roasted coffee oil is widely used in the food industry due to its characteristic and pleasant flavor.

Coffee oil yield is influenced by extraction methods, as observed for other beans, which affect the chemical composition and its applicability. The most used methods described in the scientific literature to extract coffee oil are pressing, Soxhlet, and supercritical fluid extraction. Mechanical pressing is a traditional solvent-free process, but with lower yields and oil purity, although it is mostly used in industrial facilities. Soxhlet extraction uses organic solvents and offers good efficiency and purity, although it is time-consuming and has an environmental and industrial impact due to the use of large volumes of solvent. Microwave-assisted extraction (MAE) stands out for its speed and lower solvent consumption, although it presents challenges related to extraction uniformity and equipment cost. On the other hand, extraction with supercritical fluids (SFE), especially carbon dioxide, is efficient and environmentally friendly, extracting oil with high purity and lower environmental impact, although the initial cost of the equipment is high.

Therefore, the choice of coffee oil extraction method should consider not only the yield and purity of the oil, but also aspects such as environmental impact, cost, and process time. While improved traditional techniques such as Soxhlet and mechanical pressing offer specific advantages, modern methods such as MAE and SFE offer efficiency and lower environmental impact. The decision on the most appropriate technique will depend on the specific production need and desired quality objectives.

## Figures and Tables

**Figure 1 foods-13-02601-f001:**
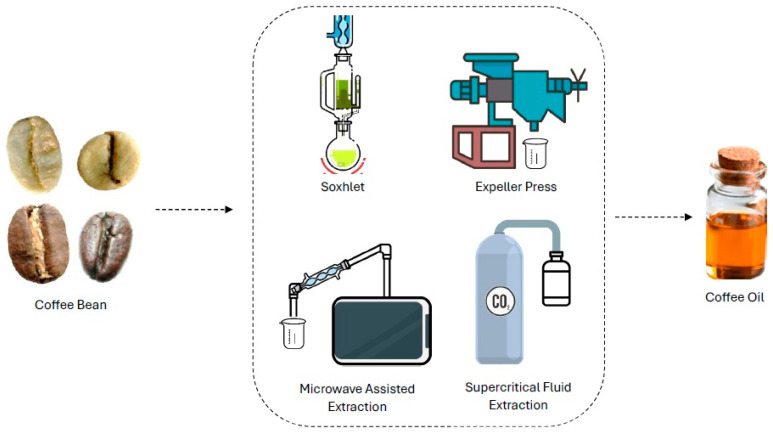
Most-used processes and techniques for coffee oil production.

**Figure 2 foods-13-02601-f002:**
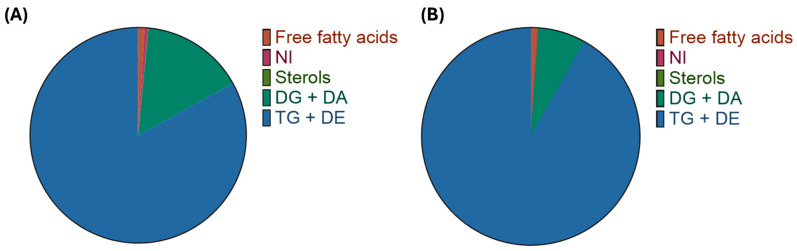
Chemical composition of Arabica (**A**) and Canephora (**B**) coffee lipid fractions (based on Lercker et al. [18]). NI: not identified; DG: diacylglycerols; DA: diterpene alcohols; TG: triacylglycerols; DE: diterpene esters.

**Figure 3 foods-13-02601-f003:**
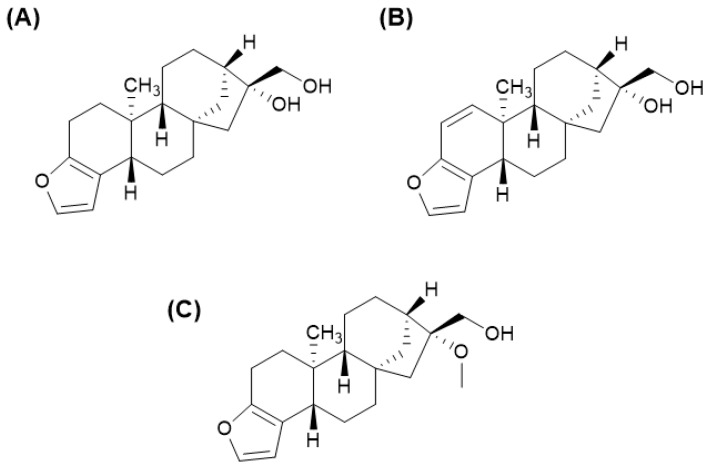
Chemical structures of: (**A**) cafestol, (**B**) kahweol, and (**C**) 16-*O*-methylcafestol.

**Figure 4 foods-13-02601-f004:**
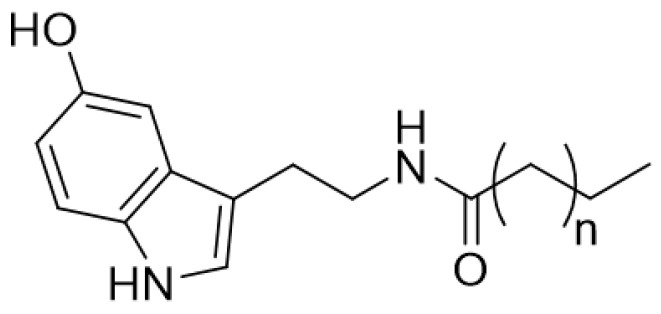
Chemical structure of ^β^*N*-alkanoyl-5-hydroxytryptamines (n = 19 or 17C, for major compounds).

## Data Availability

No new data were created or analyzed in this study. Data sharing is not applicable to this article.

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
