# Peer review of "Coffee Oil Extraction Methods: A Review"

_foods, 2024, doi:10.3390/foods13162601_

Round 1

Reviewer 1 Report

Comments and Suggestions for Authors

The document describes the chemical characteristics of coffee extracts obtained by different extraction processes. The document is well written and easy to follow. Some technical details could be incorporated for better understanding by the readers.

The authors should include the chemical composition of Coffea canephora P coffee, similar to Figure 2.

Line 131. Which coffee variety is being referred to?

Section 2.2 It will be important to include some examples of the effect of temperature on the different compounds obtained in the concept state.

Line 184-186: the authors should expand on the undesirable compounds generated during roasting. What impact do these have on the taste of the coffee, or on adverse health effects in pharmaceutical or food applications.

Section 3.1.1What is the extraction capacity of this equipment?

Sections 3.1.1 and 3.1.2 Are there data on the quality and composition of these oils?

Lines 270 and 271. The authors should indicate the optimum conditions or ranges obtained for the yield, as well as the extraction conditions suggested by the authors of the work.

The beginning of section 3.3 is missing.

It is important to include a table summarizing the main conditions employed and findings in the extraction methods.

The authors should include the work of Guatemala-Morales et al. in section 3.5.https://doi.org/10.1016/j.foodchem.2015.10.135

What are the prospects for extraction processes?

Are there any technologies that are more advantageous to the food or pharmaceutical industry?

In each section the authors should add the general advantages and disadvantages of each extraction process. If data are available, what are the costs associated with them? It would be important to include a section on applications of coffee extracts in commercial products or prototypes.

What about patented processes in the literature, as many of these studies are very small scale and their scalability is limited.

The conclusion is very general and predictable, this could be improved by focusing mainly on the extraction processes.

Author Response

Answers to Reviewer 1:

  • The authors should include the chemical composition of Coffea canephoraP coffee, similar to Figure 2.

The following paragraph has been added, including the insertion of a figure for this coffee: “Arabica and Canephora coffee beans are composed of triacylglycerols - TAG, esteri-fied ent-kaurane diterpenes, free diterpenes, sterols and their esters, tocopherols, phospho-lipids and serotonin amides[18-20]. Lipids of C. canephora var. robusta coffee were recently detailed and quantified by lipidomics through LC-ESI-MS/MS [21]. However, this study did not quantify sterols, tocopherols and diterpene esters. In order to illustrate the differ-ence in composition of Arabica and Canephora coffee lipids, Figure 2 (A and B) was done based on data from Lercker et al [18].”

  • Line 131. Which coffee variety is being referred to?

Here we were referring to Arabica coffee, which was added to the sentence.

  • Section 2.2 It will be important to include some examples of the effect of temperature on the different compounds obtained in the concept state.

This question was answered together with the subsequent question.

(4) Line 184-186: the authors should expand on the undesirable compounds generated during roasting. What impact do these have on the taste of the coffee, or on adverse health effects in pharmaceutical or food applications.

The following paragraph has been added to complement the adverse effects of roasting: “Roasted coffee oil is rich in diterpenes (3.72%), with a content of kahweol and cafestol of about 1.98 and 1.74%, respectively, depending on the roasting conditions. It also pre-sents tocopherols (0.91%), being β-tocopherol (0.88%) the main component, followed by α (0.03%) and δ isomers (0.002%). Moreover, this oil may also have chlorogenic acid (0.010%) and caffeine (0.35%), depending on the polarity of the solvent used in the extrac-tion process [53].

Roasting temperature plays a fundamental role in modifying the coffee compounds, directly influencing the quality and properties of the bean [50]. The more intense the roast, the greater the fragility of the bean wall and consequently more extravasation of the intra-cellular content occurs to the surface of the roasted bean. A recent study showed that in light roasting, there is a significant increase in the levels of cafestol and kahweol (C&K) compared to raw bean, in addition to the appearance of 7 thermal degradation products. In medium and dark roasts, 9 more compounds were observed, 10 derivatives from kahweol and 6 from cafestol [54]. Up to now, there are no studies of biological activities on these products formed by roasting.

A small amount of trans-fatty acids and peroxides are formed during coffee roasting [55]. The presence of trans-fatty acids is undesirable for food formulation due to its intake association with an increased risk of cardiovascular disease [56].  Peroxides also can negatively impact the taste of coffee, introducing bitter and metallic notes that compro-mise the sensory quality of the drink [57]. Peroxides are known for their ability to cause oxidative stress in the body, which can result in cell damage and contribute to the devel-opment of chronic diseases such as cancer and neurodegenerative diseases [58].

Another interesting study was performed by Guatemala-Morales et al. [59] related to the quantification of polycyclic aromatic hydrocarbons (PAHs) in roasted Arabica coffee. These substances have genotoxic properties and show risks to human health and, in cof-fee, are formed during the roasting of the beans. The authors validated a methodology for 16 PAHs by gas chromatography-mass spectrometry and found that a roasting in a het-erogeneous spouted bed reactor gave a sum of the PAHs ranging from 3.5 to 16.4 µg kg-1. The sum of the PAHs indicated that their concentration rise with increasing roasting temperature.”

  • Section 3.1.1 What is the extraction capacity of this equipment?

The following paragraph has been added: “In general, the oil extraction capacity of raw coffee beans using mechanical presses tends to be lower compared to roasted beans, with yields varying between 6% and 10% depending on the specific pressing conditions.”. Also, specifically in the item 3.1.1., the following text is included related to the references 63 and 73: “In this case and in the study of Vidal at al [73], it was used a laboratory screw expeller press with a capacity of 1 kg/h.” Related to the other papers in this topicv, no information about this could be found.

  • Sections 3.1.1 and 3.1.2 Are there data on the quality and composition of these oils?

The articles do not provide information on the quality or composition of the added oils. As a comparison between the methods, the following paragraph has been added: “In conclusion, mechanical pressing is a natural and straightforward approach that does not use chemical solvents, offering environmental and quality advantages when compared to solvent extraction methods that uses organic solvents, avoiding the solvent recovering step, possible problems with solvent residue and pollution, making it attractive to the food industry and pharmaceutical seeking high quality and less processed products. However, it tends to offer a lower yield and cannot achieve the same oil purity obtained by more advanced methods"

  • Lines 270 and 271. The authors should indicate the optimum conditions or ranges obtained for the yield, as well as the extraction conditions suggested by the authors of the work.

This article did not study parameters variation in the extraction process, except for the mass (g) for biomasses such as coffee cake and sediment extracts. To further demonstrate the best condition, the following paragraph was added: “The CMC-based films with the highest antioxidant capacity were obtained by incorporating green coffee oil and a 40% hydroalcoholic extract of pomace, emulsified with lecithin. As a conclusion to the work, the authors report that the films developed, especially those containing green coffee cake, appear to be a promising alternative as natural additives and for applications in active food packaging.”

  • The beginning of section 3.3 is missing.

Sequential numbering has been corrected.

  • It is important to include a table summarizing the main conditions employed and findings in the extraction methods.

We agree with the suggestion and Table S1 has been introduced in the paper. The following paragraph has been added (line 339): In an attempt to summarize the methods that will be discussed next and allow the reader to better compare the conditions used and the results obtained, Table S1 with this information can be consulted in the Supplementary Material.”

(10) The authors should include the work of Guatemala-Morales et al. in section 3.5.https://doi.org/10.1016/j.foodchem.2015.10.135

The following paragraph has been added in topic 2.2: “Another interesting study was performed by Guatemala-Morales et al. [59] related to the quantification of polycyclic aromatic hydrocarbons (PAHs) in roasted Arabica coffee. These substances have genotoxic properties and show risks to human health and, in cof-fee, are formed during the roasting of the beans. The authors validated a methodology for 16 PAHs by gas chromatography-mass spectrometry and found that a roasting in a het-erogeneous spouted bed reactor gave a sum of the PAHs ranging from 3.5 to 16.4 µg kg-1. The sum of the PAHs indicated that their concentration rise with increasing roasting temperature.”

(11) What are the prospects for extraction processes?

The following topic has been added:

“4. Prospects for extraction processes

In recent years, coffee oil extraction has seen significant advances, with a focus on process optimization and sustainability. Advanced extraction methods, such as super-critical fluid extraction, are gaining prominence as they offer greater efficiency reducing the use of organic solvents [106].

Coffee oil is also being explored in new applications, such as biodegradable films and sustainable packaging, due to its antioxidant properties, demonstrating the potential of coffee oil to improve the functionality and sustainability of packaging materials [69].

As sustainability has become a priority, there has been a growing interest in using coffee waste, such as pomace and grounds, to extract oil and other compounds. This ap-proach not only reduces waste, but also creates value from by-products [107].” 

(12) Are there any technologies that are more advantageous to the food or pharmaceutical industry?

            This question was answered together with the subsequent question.

(13) In each section the authors should add the general advantages and disadvantages of each extraction process. If data are available, what are the costs associated with them? It would be important to include a section on applications of coffee extracts in commercial products or prototypes.

At the end of each topic on extraction methods, a short discussion of the advantages and disadvantages of the present method has been added. The following paragraphs were added: " In conclusion, mechanical pressing is a natural and straightforward approach that does not use chemical solvents, offering environmental and quality advantages when compared to solvent extraction methods that uses organic solvents, avoiding the solvent recovering step, possible problems with solvent residue and pollution, making it attractive to the food industry and pharmaceutical seeking high quality and less processed prod-ucts. However, it tends to offer a lower yield and cannot achieve the same oil purity ob-tained by more advanced methods [70,71].”; “Soxhlet comparison to other extraction techniques usually has disadvantages such as the long extraction time (4-16 h) and the large amount of solvent used, producing more waste. Another disadvantage is the exposure of the extracted solute to the solvent boiling point for a long period of time and the possibility of heating thermolabile compounds, bringing undesirable results [86-88].” and " Therefore, although it requires a high cost in terms of equipment and complex pro-cessing, SFE makes it possible to obtain a high purity oil, without solvent residues, and is a highly efficient and sustainable method, preserving the bioactive compounds, offering advantages for food and pharmaceutical industry that requires high oil quality [102].”

(14) What about patented processes in the literature, as many of these studies are very small scale and their scalability is limited.

This review article consists of scientific prospecting and does not deal with technological prospecting. However, in order to cite this relevant data, an in-depth search in the WIPO patent database was carried out and the following paragraph was added to topic 3.4: " To establish a basis on the generation of related patents, a preliminary search was conducted in the World Intellectual Property Organization (WIPO) database using the terms: "coffee oil" AND "extraction". This research resulted in the identification of 61 documents, of which 16 are associated with the United Kingdom, 8 with Japan and the United States, 5 with the Russian Federation, 4 with Canada and the Republic of Korea, 3 with New Zealand and the Cooperation Treaty in Patents (PCT), and 2 to China and the European Patent Office. The documents were published between 2015 and 2024. However, given that these patents are not the main focus of this work, they will not be analyzed in greater depth."

(15) The conclusion is very general and predictable, this could be improved by focusing mainly on the extraction processes.

The following paragraph has been added in conclusion: “Coffee oil yield is influenced by extraction methods, as observed for other beans, which affect the chemical composition and its applicability. The most used methods de-scribed in the scientific literature to extract coffee oil are pressing, Soxhlet and supercritical fluid extraction. Mechanical pressing is a traditional solvent-free process, but with lower yields and oil purity, although mostly used in the industrial facilities. Soxhlet extraction uses organic solvents and offers good efficiency and purity, although it is time-consuming and has an environmental and industrial impact due to the use of large volumes of sol-vent. Microwave-assisted extraction (MAE) stands out for its speed and lower solvent consumption, although it presents challenges related to extraction uniformity and equip-ment cost. On the other hand, extraction with supercritical fluids (SFE), especially carbon dioxide, is efficient and environmentally friendly, extracting oil with high purity and low-er environmental impact, although the initial cost of the equipment is high.

Therefore, the choice of coffee oil extraction method should consider not only the yield and purity of the oil, but also aspects such as environmental impact, cost and pro-cess time. While improved traditional techniques such as Soxhlet and mechanical press-ing offer specific advantages, modern methods such as MAE and SFE are offer efficiency and lower environmental impact. The decision on the most appropriate technique will depend on the specific production need and desired quality objectives.”

Reviewer 2 Report

Comments and Suggestions for Authors

Please consider to remove or complement the text between lines 103-105, given this information is repeated in Figure 2.

Please revise the molecular structures of cafestol and kahweol showed in Figure 3. Also, please increase the resolution of the image.

Please consider to include a Table to summarize and compare the different methods currently used for coffee oil extraction, including operating conditions, cost, extraction yield, advantages and main issues for each method as well as quality indicators (for instance, important chemical compounds) of the oil extracted.

Comments on the Quality of English Language

Please revise English grammar, given there are some instances which require minor amendments, such as:

L55. ... as important sources of ...

L96. ... are the techniques ...

L105. ... Lipids of C. ...

L106. ... coffee were recently ...

L107. ... verified that ...

L147. ... a fermentative ...

Author Response

Answers to Reviewer 2:

  • Please consider to remove or complement the text between lines 103-105, given this information is repeated in Figure 2.

The text has been altered, removing the written percentages and leaving only the image, as can be seen: “Arabica and Canephora coffee beans are composed of triacylglycerols - TAG, esterified ent-kaurane diterpenes, free diterpenes, sterols and their esters, tocopherols, phospho-lipids and serotonin amides [18-20].”

  • Please revise the molecular structures of cafestol and kahweol showed in Figure 3. Also, please increase the resolution of the image.

 The structures represented in Figure 3 have been changed.

  • Please consider to include a Table to summarize and compare the different methods currently used for coffee oil extraction, including operating conditions, cost, extraction yield, advantages and main issues for each method as well as quality indicators (for instance, important chemical compounds) of the oil extracted.

The advantages of the extraction methods were included in the text, at the end of each topic. A table summarizing the main conditions used and the results of the extraction methods has been added as supplementary material (Table S1) and described in the text.

  • Please revise English grammar, given there are some instances which require minor amendments, such as:

L55. ... as important sources of ...

L96. ... are the techniques ...

L105. ... Lipids of C. ...

L106. ... coffee were recently ...

L107. ... verified that ...

L147. ... a fermentative ...

Thank you for your attention to detail. The minor adjustments have been made.

Reviewer 3 Report

Comments and Suggestions for Authors

Correct the citation template line 101, 133

Lines 58-59 no explained abbreviations PLP, LLL etc.

Lines 101 -102 repeated information from lines 53-54

In the caption under Figure 2 add source data - literature (as in the text describing the content of individual compounds)

No names of individual acids in lines 132-133

Line 142 add: presented in Figure 4,

Line 240: Due to the low oil recovery through the hydraulic press, this method is not very popular in coffee oil extraction. In the summary, the authors write: The most used methods to extract coffee oil are pressing, … which is contrary to the previous statement.

Line 515 pressurized liquid extraction showed the highest tocopherol, total phenols and phytosterol content, However, the authors do not describe this method.

Point 3.2 - difficult to read, lack of consistency in the description of the main methods, which are referred to in different places. Some of the data given for these methods is too detailed and therefore difficult to compare.

Other methods (point 3.5) are practically not described at all, apart from giving the efficiency and main components.

References 13 - authors' names entered at the end of the line.

Author Response

Answers to Reviewer 3:

  • Correct the citation template line 101, 133.

The citation template for lines 101 and 133 has been corrected.

  • Lines 58-59 no explained abbreviations PLP, LLL etc.

Abbreviations have been replaced by nomenclature.

  • Lines 101 -102 repeated information from lines 53-54

The information that the main class of compounds in coffee oil are triacyl glycerides (up to 75%) is found in two different places because first we present triacyl glycerides specifically (lines 53-54), emphasizing what they are, and then we present them together with the other components (lines 101-102), in graph form for better visualization. In the second appearance, the percentages were removed from the text and left only in the figure.

  • In the caption under Figure 2 add source data - literature (as in the text describing the content of individual compounds

The references have been added to the legend in Figure 2.

  • No names of individual acids in lines 132-133

The nomenclatures have been added after each acid, in parentheses.

  • Line 142 add: presented in Figure 4

The information was added.

  • Line 240: Due to the low oil recovery through the hydraulic press, this method is not very popular in coffee oil extraction. In the summary, the authors write: The most used methods to extract coffee oil are pressing, … which is contrary to the previous statement

One of the most widely used methods for extracting coffee oil is expeller pressing. Since hydraulic pressing is not a common processing method, its topic was removed and the article that uses this method was added to topic 3.5, being: “A mechanical hydraulic press, much less used than expeller press, …”

  • Line 515 pressurized liquid extraction showed the highest tocopherol, total phenols and phytosterol content, However, the authors do not describe this method.

The submitted work aimed to investigate the coffee oil extraction techniques available in the literature. Therefore, all the articles referred to were summarized with a focus on the oil extraction processes and their consequent percentage results, presenting secondarily the other results investigated, specific to each article. In this example of the article by Dong et al., the methods for obtaining each compound were not evidenced, following the same parameter as the other articles addressed during the work.

  • Point 3.2 - difficult to read, lack of consistency in the description of the main methods, which are referred to in different places. Some of the data given for these methods is too detailed and therefore difficult to compare.

Topic 3.2 has been reorganized and completed. A table in the supplementary material (Table S1) was also added to better compare results.

  • Other methods (point 3.5) are practically not described at all, apart from giving the efficiency and main components.

The work of Dong et al. [105] was augmented with: “The ultrasound technique consisted of using a working frequency of 40 kHz and electrical power of 50 W, at a sample:ethanol ratio of 1:30 (g/mL) at 35 °C for 50 min. The microwave technique consisted of using 10.0 g of powdered sample with 100 mL of ethanol and then extracted at 60 °C for 30 min, and the microwave power was set at 200 W. The ultra-sound-microwave-assisted extraction used a mass ratio with ethanol of 1:28 (g/mL) at 60 °C for 10 min, with a microwave power of 350 W. The last technique, pressurized liquid extraction, consisted of using a sample to ethanol ratio of 1:2 (g/mL) at 100 °C for 30 min, and the pressure reached 100 bar. Among the four techniques evaluated, the ultrasonic/microwave-assisted extraction presented the highest yield (10.58%), followed by microwave-assisted extraction (9.34%), ultrasonic-assisted extraction (9.06%) and pressurized liquid extraction (6.34%).”

Techniques other than those described in the topics above have not yet been widely explored by researchers in the field.

  • References 13 - authors' names entered at the end of the line.

We appreciate the attention to detail and this reference has been fixed.